# The Preventable Effect of Taekwondo Sport among Cadets and Junior’ Bone Mineral Density: DEXA Assessment

**DOI:** 10.3390/children10010170

**Published:** 2023-01-15

**Authors:** Hadeel Ali Ghazzawi, Adam Tawfiq Amawi, Hamza Alduraidi, Malik Juweid, Hussam H. Alhawari, Mousa A. Al-Abbadi, Ali M. Alabbadi, Lana Salah Subhi AlNemer

**Affiliations:** 1Department Nutrition and Food Technology, School of Agriculture, The University of Jordan, Amman 11942, Jordan; 2Department of Physical and Health Education, Faculty of Educational Sciences, Al-Ahliyya Amman University, Al-Salt 19328, Jordan; 3Community Health Nursing Department, School of Nursing, The University of Jordan, Amman 11942, Jordan; 4Department of Radiology and Nuclear Medicine, School of Medicine, The University of Jordan, Amman 11942, Jordan; 5Department of Internal Medicine, School of Medicine, The University of Jordan, Amman 11942, Jordan; 6Department of Pathology, Microbiology and Forensic Medicine, School of Medicine, The University of Jordan, Amman 11942, Jordan

**Keywords:** weight category, rapid weight loss, children, bone mass density, DEXA, adolescence, Taekwondo

## Abstract

Athletes competing in Taekwondo (TKD), the weight-category sport, tend to rapidly lose weight to achieve the desired body weight for better competitive results. Little is known about the effect of rapid weight reduction on bone mass density (BMD), especially during childhood and adolescence. The current study aimed to investigate the impact of rapid weight loss on BMD among cadets and juniors TKD athletes. A descriptive case series study design was conducted and collected from 28 males and females aged 12–17 years old, with mean age 14.4 ± 1.7. Dual-energy X-ray absorptiometry (DEXA) was used for both BMD and body composition assessment, and laboratory tests were also performed for the total calcium (Ca), TSH, free T4 (FT4), and 25-OH-vitamin D. Results showed normal levels of Ca (82.1%), TSH (96.4%), and FT4 (96.4%), whilst 85.7% had vitamin D deficiency. DEXA results showed that within male athletes, juniors had a wider range of BMD than cadets, while within females, results did not vary, with no statistical difference between both males and females. Our results suggested that children and adolescents’ BMD was positively related to TKD sport regardless of the abnormal weight loss strategies used, as evidenced by laboratory results. Children and adolescents should be conscious and practice TKD sport adopting healthy weight loss behaviors.

## 1. Introduction

Taekwondo, one of the Olympic combat sports, is a Korean martial art characterized by fast kicking techniques, jumping, and spinning kicks, where body weight is an important concern [1,2]. Officially, athletes that participate in weight-category sports such as Taekwondo, which is maintained by a weigh-in procedure the day before competition, tend to excessively and rapidly lose weight with a range of 2–10% to compete in a lighter weight class [1]. Athletic performance is highly affected by adjusting body composition and managing body weight [3]. Generally, according to the athletes’ belief, achieving a lower body weight has beneficial competitive results and has been evidenced to indicate a sense of “sporting identity” [3,4].

Several rapid and controversial weight loss strategies are widely used by athletes who compete in weight-category sports and undergo weighing prior to competitions. Food restriction and dehydration are known to be the most common weight loss strategies, leading to a significant loss in body mass within days and weeks [3,4]. These strategies inversely affect athletes’ health, causing hormonal fluctuations, psychological changes, electrolyte imbalance, and impaired immune functions [5]. Athletes in weight-category sports undertake several other unconventional practices, including fasting, skipping meals, use of diuretics, laxatives, and diet pills, excessively exercising, sauna use, and training with rubber suits [6]. Rapid weight loss has harmful impacts on health as it causes lean mass loss (muscles and body water), as well as loss of gastrointestinal contents [3]. Consequently, glycogen availability and body fluid are both altered [4]. Strictly, among combat sport athletes, dehydration is mainly caused by three factors, including fluid restriction, releasing bound body fluids such as expending glycogen stores, and wasting fluids such as via sweating, urination, and respiration [5]. Indeed, dehydration is a health hazard leading to a reduction in plasma volume, and increasing the heart rate, while the difference in arteriovenous oxygen is lowered during exercise [4]. Nonetheless, these weight loss strategies and practices may lead to significant health problems, including impaired growth and development, diminished physical performance, and compromised nutritional status [6].

Normally, childhood and adolescence are the critical period of the human lifecycle when pubescence growth and bone development occur, increasing energy and nutrient demands along with psychological and social changes [6,7]. During this key stage, it is important to ensure an adequate nutrient intake to maintain health, normal growth, and maturation, thus optimizing performance. Furthermore, nutrition is challenging for adolescent athletes, especially when participating in weight-category sports [7]. Practicing such harmful strategies to achieve rapid weight loss is hazardous, especially during adolescence [5]. Hence, this age group is particularly at high risk when adopting such practices. The reduction in food consumption inversely affects anabolic processes, leading to an increase in inflammatory mediators and a reduction in IGF-1 secretion. Some results have shown cases of death related to extreme and rapid weight loss [6]. In addition, calcium dietary requirements increase to maintain optimal bone growth, development, and calcium balance, and thus consumption up to 1500 mg/day is recommended [7]. Along with calcium, vitamin D is an essential nutrient that has gained growing attention regarding sport nutrition. In relation with calcium, vitamin D is necessary for its homeostasis and regulation, noting that vitamin D is critical for bone mineralization, maintaining skeletal integrity, and is effective in preventing secondary hyperparathyroidism. It is recommended for athletes’ serum 25 (OH)D levels to be less than 32 ng/mL [8,9]. Recently, vitamin D deficiency has been discovered to be prevalent among Jordanian males (30.97%) and females (43.01%), with a significant difference among age groups, showing a prevalence of 41.17% among children with an age range between 1 and 14 years, whereas the adult prevalence was 40.05% for ages of 15 to 83 years [10].

Bone is a tissue that is metabolically active and continuously undergoes remodeling throughout the lifecycle, which during the second and third decades, its peak mass is attained, reaching 90% [11,12]. To avoid fracture diseases such as osteoporosis in late stages of life and adulthood, gaining bone mass during childhood and adolescence periods is vital. Amongst the multiple factors affecting BMD acquisition, vitamin D and calcium consumption, psychological factors, being physically active, and being a member of a sport team are the most important factors that affect BMD health [12]. Moreover, bone density is affected by sports type in different levels: weight-category sports are considered effective and preventable strategies of osteoporosis as it is known that bone development largely depends on mechanical loading [11,13]. Therefore, in the evaluation of bone mass during the pediatric period, various imaging procedures are used, with DEXA being one of the most commonly used safe techniques [12]. Currently, dual-energy X-ray absorptiometry (DEXA) is the golden standard for bone mineral density quantification, recommended by the World Health Organization (WHO) and widely employed for quantitative radiologic bone mass assessment. DEXA is simple, available, and the exposure to radiation is relatively low [13,14].

Despite the large number of studies investigating the impact of martial art sports on bone health, little is known regarding the association between the weight reduction cycle adopted by Taekwondo athletes and their bone density during the crucial period of childhood and adolescence, particularly in Jordan. Thus, the present study aims to evaluate the effect of rapid weight loss on bone density among Jordanian adolescent cadets and juniors aged between 12 and 17 years who are participating in Taekwondo using DEXA assessment.

## 2. Materials and Methods

A descriptive case series study design consisting of anthropometric measurements, DEXA, and blood biomarkers’ sampling was used. Forty elite Jordanian athletes were planned to be recruited from the national team in Jordan: Twenty cadet players (age 12–14) from all weight classesTwenty junior players (age 15–17) from all weight classes

All athletes filled out the informed consent form to participate in this study. Only 28 athletes participated in the study as we excluded the higher weight categories. Each athletes attended The National Center for Diabetes Endocrinology and Genetics (NCDEG) authorized by the University of Jordan in Amman to undertake the DEXA procedure on one occasion. On this visit, anthropometric measurements were obtained by the research assistant. 

Cadets:

Born in 2007, 2008, and 2009 to be able to compete in 2021/2022 under the cadets’ age category. 

Cadet male weight classes: −33 kg, −37 kg, −41 kg, −45 kg, −49 kg, −53 kg, −57 kg, −61 kg, −65 kg, +65 kg.

Cadet female weight classes: −29 kg, −33 kg, −37 kg, −41 kg, −44 kg, −47 kg, −51 kg, −55 kg, −59 kg, +59 kg.

Juniors: 

Born in 2004, 2005, and 2006 to be able to compete in 2021/2022 under the cadets’ age category.

Junior male weight classes: −45 kg, −48 kg, −51 kg, −55 kg, −59 kg, −63 kg, −68 kg, −73 kg, −78 kg, +78 kg.

Junior female weight classes: −42 kg, −44 kg, −46 kg, −49 kg, −52 kg, −55 kg, −59 kg, −63 kg, −68 kg, +68 kg.

Inclusion criteria included: elite TKD cadets and junior black-belt athletes who are actively competing at the national-team level or more prominent competitions (National/Kingdom medal winners, Regional/International medal winners), aged from 12 to 17 years old, males and females, within the weight classes.

To describe the BMD measurements, they were first converted into Z-scores. Z-scores are similar to T-scores except that instead of comparing the patient’s BMD with the young adult mean, it is compared with the mean BMD expected for the patient’s peers (for example, for a healthy normal subject matched for age, gender, and ethnic group, which were already calculated in the DEXA scan software). Therefore, it was not necessary to recruit a control group for comparing the participants’ results. Hence, the reference values of the DEXA scan compared the results with the matched age and sex from a healthy population. Over the last 10 years, the interpretation of DEXA scans has been guided by the WHO T-score definition of bone diseases [15]. 

Exclusion criteria included: age under 12 or over 17, former athletes, and has had a bone injury or fracture in the previous 6 months. 

Research instrument: A DEXA scanner [16] was used for the assessment of both BMD and body composition at the NCDEG—National Center for Diabetes, Endocrinology, and Genetic. Biochemical assessment of the calcium levels was performed utilizing the photometric absorbance method (Cobas C311/501, test reference number 05061482, by Roche Diagnostics, Indianapolis, IN, USA). The TSH measurement was performed by a chemiluminescent immunoassay (Adivia Centaur, Siemens Diagnostics, Deerfield, IL, USA), while the vitamin D measurement was performed utilizing the chemiluminescent microparticle immunoassay (Alinity, Abbott, Abbott Park, IL, USA).

Baseline BMD was measured using a dual-energy X-ray absorptiometer (DEXA) (lunar iDEXA-GE, Madison, WI, USA), which included lumbar spine (i.e., L1 to L4) and femoral neck. Lumbar mineral density was calculated as: baseline vs. postoperative results. Characteristics of the preoperative DEXA scan (*n* = 21), postoperative DEXA scan (*n* = 21), mean difference *p*-value, and overall lumbar BMD, g/cm^2^, were 1.00 ± 0.23, 1.04 ± 0.21, 0.04, 0.001 *, for males: 1.00 ± 0.23, 1.03 ± 0.21, 0.03, 0.027 *, pediatric males: 0.80 ± 0.25, 0.86 ± 0.23, 0.06, 0.091, and adult males: 1.13 ± 0.1, 1.15 ± 0.11, 0.02, 0.165. For females: 1.01 ± 0.25, 1.05 ± 0.23, 0.04, 0.014 *, pediatric females: 0.88 ± 0.49, 0.93 ± 0.48, 0.05, 0.101, and for adult females: 1.06 ± 0.16, 1.09 ± 0.14, 0.03, 0.082. Overall lumbar Z-score: −0.97 ± 1.17, −0.77 ± 1.14, 0.20, 0.026 *, for males: −1.05 ± 0.89, −0.90 ± 0.82, 0.15, 0.102, pediatric males: −1.58 ± 1.08, −1.47 ± 0.79, 0.11, 0.395, and adult males: −0.72 ± 0.62, −0.54 ± 0.64, 0.18, 0.143. For females: −0.84 ± 1.57, −0.58 ± 1.57, 0.26, 0.157, pediatric females: −0.43 ± 1.97, −0.02 ± 2.4, 0.41, 0.395, and adult females: −0.98 ± 1.61, −0.77 ± 0.15, 0.21, 0.355. Data are presented as number (%) or mean ± standard deviation (SD). Asterisks (*) represent statistically significant results. Figure 1 presents pre-operative vs. post-operative lumbar and femoral bone mineral density (g/cm^2^ and Z-score). A second scan was performed between six months and one year after surgery. The DEXA machine was calibrated daily using a standard phantom calibrator provided by the manufacturer. Measurements were maintained within precision standards of 1.0%. The least significant change (LSC) for lumbar spine (L1–L4) BMD at our institution was 0.028 g/cm^2^, while the LSC for the left femur BMD was 0.033 g/cm^2^. All scans were performed by trained technicians and analyzed by an experienced radiologist. Quality control procedures were followed in accordance with the manufacturer’s recommendations. Height was measured without shoes to the nearest 0.1 cm using a fixed stadiometer. Body weight (kilograms) was measured to the nearest 0.1 kg with the participant in light clothing without shoes. Body mass index (BMI) was defined as the weight in kilograms divided by the square of the height in meters.

### 2.1. Statistical Analysis

Statistical analysis was performed by using the Statistical Package for the Social Sciences (SPSS) software, release 25.0 (SPSS, Inc., Chicago, IL, USA). Descriptive analysis was conducted to determine means and standard deviation (SD) for quantitative variables. Statistical significance was defined as *p* < 0.05. Univariate regression and one-way ANOVA models were used to determine the impact of each weight class on the nutritional intake routine and weight loss strategy. The ANOVA models included independent determinants to predict weight class, BMD, nutritional routine, and weight loss protocols with gender, age, medal during the year, age of enrolments, age of starting competition, body composition, energy-adjusted dietary intakes, and supplements’ intake.

A comparison with a control group was not attainable at this stage because some of the variables that need to be collected require time-consuming laboratory and radiology work. We, however, recommend future studies to consider comparing with non-athletic controls. Due to the relatively small sample size of the study, determining the 95th percentile would not work ideally.

### 2.2. Ethical Approval

The Institutional Review Board at The University of Jordan (IRB at UJ) approved the research proposal submitted by Dr. Hadeel Ali Ghazzawi from the School of Agriculture, Decision No. (37-2022): The IRB at The University of Jordan.

## 3. Results

### 3.1. Sample Characteristics

A sample of 28 Jordanian TKD athletes was investigated in this study, consisting of 15 males (53.6%) and 13 (46.4%) females. Age ranged from 12 to 17 years, with a mean of 14.4 (± 1.7) years. The athletes were ranked according to their age into 11 (39.3%) cadets, aged up to 14 years, and 17 (60.7%) juniors, aged 15 years or more. The athletes’ weights ranged from 33 to 92 kg, with a mean of 54.1 (±13.9) kg, whereas heights ranged from 1.4 to 1.9 meters, with a mean of 1.65 (±0.12) meters, and body mass index (BMI) ranged from 15.3 to 26.8, with a mean of 19.4 (±2.7).

Within the cadet group, 5 (45.5%) were males who had a mean age of 13.2 (±1.5) years, a mean weight of 44.8 (± 9.7) kg, a mean height of 1.60 (±0.11) meters, and a mean BMI of 17.4 (±1.7). Six (54.5%) were females, who had a mean age of 12.4 (±0.9) years, a mean weight of 45.3 (± 8.3) kg, a mean height of 1.55 (±0.07) meters, and a mean BMI of 18.7 (±2.5). 

Within the junior group, ten (58.8%) were males who had a mean age of 15.3 (±0.9) years, a mean weight of 63.3 (±15.3) kg, a mean height of 1.74 (±0.12) meters, and a mean BMI of 20.6 (±3.1). Seven (41.2%) were females, who had a mean age of 15.8 (±0.9) years, a mean weight of 54.9 (±10.2) kg, a mean height of 1.66 (±0.08) meters, and a mean BMI of 20.0 (± 2.4).

The mean age, weight, height, and BMI of all males (*n* = 15), all females (*n* = 13), all cadets (*n* = 11), and all juniors (*n* = 17) are displayed in Table 1, along with their respective standard deviations.

### 3.2. Laboratory Results

Sample members were tested in the laboratory for their levels of total calcium, TSH, free T4 (FT4), and 25-OH-vitamin D. As shown in Table 2, most athletes showed a normal level of calcium (Ca) (82.1%), TSH (96.4%), and FT4 (96.4%), whereas none of the athletes showed a sufficient level of vitamin D, with 14.3% having insufficiency, and an alarming 85.7% having deficiency of vitamin D. No significant differences were found between males and females, or between cadets and juniors in terms of lab results’ categories.

### 3.3. Biological Parameters

Sample members were also measured in terms of biological parameters, including tissue fat, region fat, tissue gram, fat gram, lean gram, BMC gram, and kg in the body parts of arms, legs, trunk, android, and gynoid, as well as total body, as shown in Table 3. Furthermore, an independent-sample t test was used to compare the mean of total body biological parameters between male and female athletes, where tissue fat (*p* < 0.001) and region fat (*p* < 0.001) were significantly lower in males than females, while lean gram (*p* = 0.009) was significantly higher in males than females. The remaining total body biological measures did not vary significantly based on gender (Table 3).

Furthermore, BMD was assessed in the laboratory using the Z-score among cadets and juniors in male and female athlete groups. Figure 1 shows that within male athletes, juniors tend to have a wider range of BMD than cadets, with a tendency towards lower Z-scores in spine, femoral neck, and total body. Within female athletes, cadets and juniors did not vary a lot in their Z-score range, but juniors had a tendency towards higher Z-scores. These differences between male and female athletes were tested for statistical significance using an independent-sample t test, but none of the comparisons showed a statistically significant difference in their Z-score means. Additionally, cadets and juniors were compared in terms of their mean Z-score, where none of the comparisons showed a statistically significant difference.

Figure 2 shows the comparison between children and adolescents with normal bone mineral density who are not Taekwondo or other martial arts athletes. No significant differences were found when Z-scores were compared among cadets, juniors, and controls in any of the areas (lumber 1–4, femoral neck, or total body).

However, other sample characteristics were significantly correlated to Z-score, such as weight, height, and BMI, but age and laboratory results were not found significantly correlated to Z-score (Table 4).

## 4. Discussion

The present study’s purpose was to evaluate the effect of weight category on bone density among a sample of children and adolescent Jordanian Taekwondo athletes, who were categorized into cadets and juniors. Our results showed that both cadets and juniors had normal levels of calcium (Ca) (82.1%), TSH (96.4%), and FT4 (96.4%), while vitamin D was deficient with an alarming and warning value (85.7%), and with no statistical difference between both genders.

Hence, a reduction in calcium absorption occurs when serum vitamin D is low, resulting in elevated parathyroid hormone levels; consequently, this causes osteoclasts’ activation, which leads to bone collagen break down [9]. Moreover, elevated secretion of parathyroid hormone (PTH), decreased BMD, and increased risk of bone fracture is linked to low levels of serum 25 (OH)D [8]. Likewise, Millward et al. reported a 12% higher prevalence of stress fractures among athletes with vitamin D deficiency compared to those with normal levels, showing that restoring vitamin D levels is important in reducing the risk of stress fractures among athletes [17]. A positive association between serum vitamin D levels and mean power output and relative mean power output was found among Taekwondo athletes aged 15–18 years old [18].

Thus, correcting vitamin D deficiency is a must, which can be achieved by adopting healthy life habits to correct vitamin D levels, which is fundamental for optimizing athletic performance and maintaining healthy muscles. High-intensity exercises lead to oxidative stress, which can be effectively reduced by vitamin D [9]. Hence, physical activity and calcium supplements positively affect BMD among adolescents as calcium is beneficial in enhancing BMD and reducing the risk of fractures [19,20].

Furthermore, regarding biological parameters, fat tissue was significantly higher among females compared with males, with a *p*-value less than 0.005, whilst lean gram was significantly higher in males than females, with a *p*-value of 0.009. Surprisingly, we found a wider range of BMD among juniors than cadets within males, whereas within females no variation between cadets and juniors was observed, as well as no statistically significant difference between both genders, noting that BMD ranges among participants were relatively normal. This could be simply explained by the beneficial effect of an active lifestyle during this critical period. Likewise, Han et al. conducted a study among adolescents aged between 12 and 18 years old and found that BMD was higher among adolescents who were physically active and spent more time outdoors [19]. Moreover, Hong and Kim suggested that bone health is more significantly affected by weight-bearing and resistance exercises. These types of sports are highly affected by gravity [21].

In addition, Shin et al. identified the impact of Taekwondo training on bone health and assumed that among the group competing in Taekwondo, the average BMD was greater compared to a sedentary control group. This study included a sample of Korean female students aged 13–17 years old, suggesting that during growth, adolescent females should practice a weight-category sport such as Taekwondo [22,23]. 

Indeed, practicing moderate- to high-intensity exercises for 60 minutes per day during childhood and adolescence is recommended by the Centers for Disease Control and Prevention (CDC). Both dietary intake and physical activity were previously proven to be essential for bone health [21,23].

On the other hand, in a recent study conducted by Norsuriani and Ooi, no statistical difference was found between Taekwondo, Silat, and sedentary groups in a quantitative ultrasound measurement of the bone speed of sound (SOS) of dominant and non-dominant arms and legs when examining bone health status in Malay female adolescents, with an age range between 15 and 19 years old [11].

Although Taekwondo is a weight-category sport, and athletes fighting in such sports undergo several abnormal eating behaviors, which are known to be harmful especially during adolescence [5], we found no negative impact on BMD. It is clear that bone density is affected by multiple factors, such as an active lifestyle, vitamin D and calcium intake, and other psychological factors [12].

The current study has many strengths, which is novelity starts from focusing on a recent and important public issue among athletes during key and critical periods throughout the lifecycle. The results of this research encourage adolescent athletes, especially those competing in weight-category sports including Taekwondo, to adopt healthy lifestyle habits to maintain optimal bone density and maximize physical performance. DEXA was used to fulfill our objective, which is highly recommended by the WHO and widely used. In addition, a significant association was found using a powerful statistical analysis, which yielded strong results. Nevertheless, this study is limited by the small sample size.

## 5. Conclusions

In conclusion, regardless of the hazardous health effects caused by the rapid weight cycle adopted by Taekwondo athletes, we found that BMD was positively associated with Taekwondo, showing a beneficial effect on bone health. Additionally, Taekwondo and exercise showed a potential protective effect for bone density. Thus, engaging in such type of sport effectively enhances bone health. Consequently, it is important for children and adolescents to practice high-intensity exercises, especially in this critical period, to improve their bone health, to maximize performance, and to achieve better competition results. Nevertheless, nutrient intake should be carefully maintained, and junior athletes should be aware of their dietary habits and food intake. 

## Figures and Tables

**Figure 1 children-10-00170-f001:**
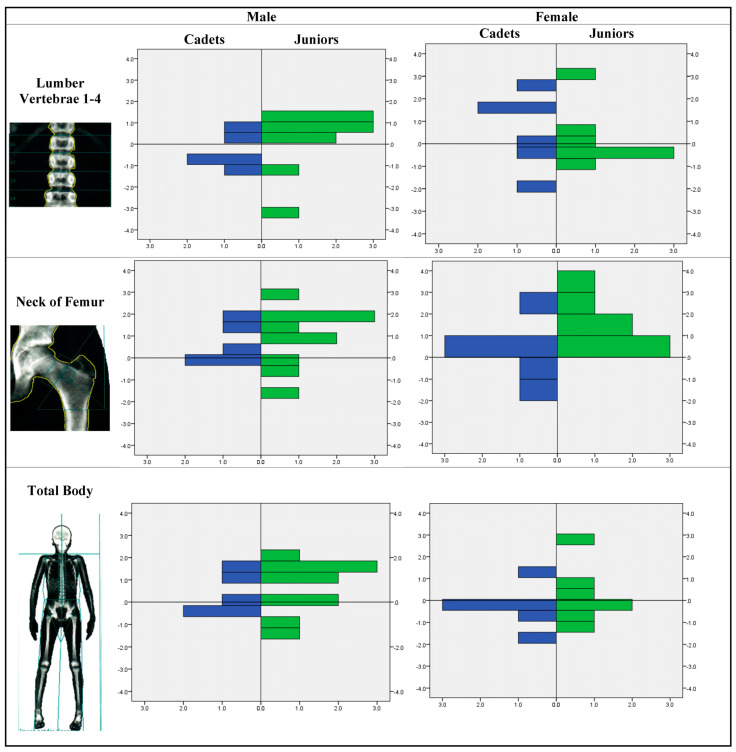
Mineral density Z-score distribution in lumber 1–4, femoral neck, and total.

**Figure 2 children-10-00170-f002:**
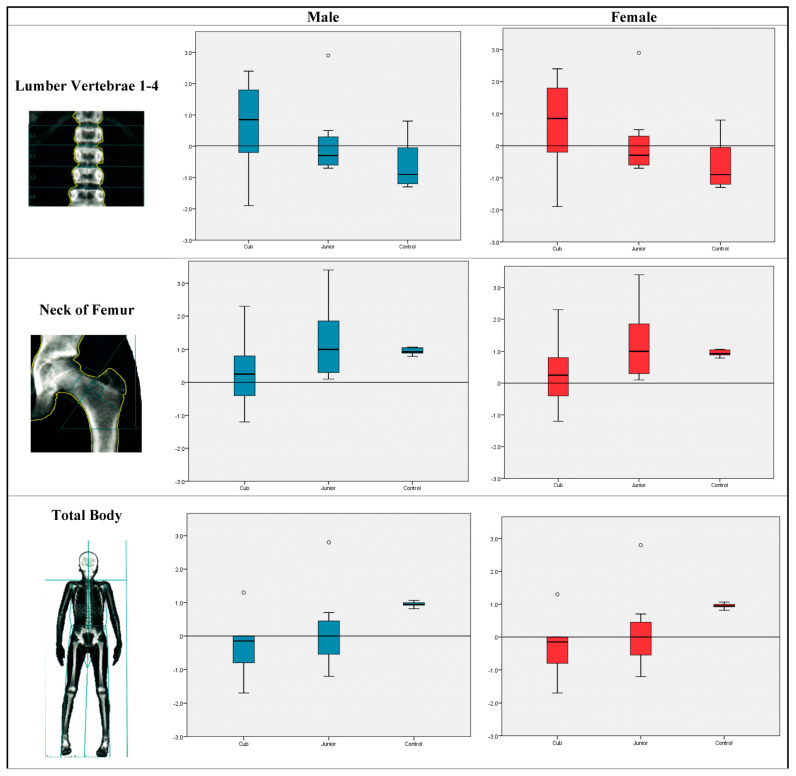
Mineral density Z-score distribution in lumber 1–4, femoral neck, and total body among cadets, juniors, and non-athletic controls.

**Table 1 children-10-00170-t001:** Sample characteristics (*n* = 28).

	Rank	Cadets (*n* = 11)		Juniors (*n* = 17)		Total	
Gender		Mean	SD	Mean	SD	Mean	SD
**Male** **(*n* = 15)**	Age (years)	13.2	1.5	15.3	0.9	14.6	1.5
	Weight (kg)	44.8	9.7	63.3	15.3	57.1	16.1
	Height (m)	1.60	0.11	1.74	0.12	1.69	0.13
	BMI	17.4	1.7	20.6	3.1	19.5	3.1
**Female** **(*n* = 13)**	Age (years)	12.4	0.9	15.8	0.9	14.2	2.0
	Weight (kg)	45.3	8.3	54.9	10.2	50.5	10.3
	Height (m)	1.55	0.07	1.66	0.08	1.61	0.09
	BMI	18.7	2.5	20.0	2.4	19.4	2.41
**Total**	Age (years)	12.7	1.2	15.5	0.9	14.4	1.7
	Weight (kg)	45.1	8.5	59.8	13.7	54.1	13.9
	Height (m)	1.57	0.09	1.71	0.11	1.65	0.12
	BMI	18.1	2.2	20.3	2.7	19.4	2.7

BMI: body mass index.

**Table 2 children-10-00170-t002:** Laboratory levels and categories of Ca, TSH, FT4, and Vit. D (*n* = 28).

Lab. Test	Mean (SD)	Categories	Count (%)
Ca. (mg/dL)	9.93 (0.34)	Normal	23 (82.1)
		Hypercalcemia	5 (17.9)
TSH-3 (mIU/mL)	1.46 (0.73)	Normal	27 (96.4)
		Elevated TSH-3	1 (3.6)
FT4 ( pmol/L)	15.35 (2.51)	Normal	27 (96.4)
		Elevated FT4	1 (3.6)
Vit. D (ng/mL)	13.13 (5.40)	Sufficient	0 (0)
		Insufficient	4 (14.3)
		Deficient	24 (85.7)

**Table 3 children-10-00170-t003:** Biological parameters’ means and their gender-based comparisons (*n* = 28).

Parameter	Body Part, Mean					Total, Mean	Total Male vs. Female	
	Arms	Legs	Trunk	Android	Gynoid		t	*p*
Tissue Fat	23	25	19	16	25	22	−4.941	0.000 *
Region Fat	22	24	19	16	24	21	−4.924	0.000 *
Tissue Gram	5852	18,923	23,804	3082	8265	52,350	1.413	0.170
Fat Gram	1349	4668	4702	541	2076	11,513	−1.422	0.167
Lean Gram	4503	14,255	19,102	2541	6189	40,838	2.830	0.009 *
BMC Gram	282	882	639	40	231	2181	1.646	0.113
kg	6	20	24	3	8	54	1.425	0.166

* Statistically significant, *p* < 0.05.

**Table 4 children-10-00170-t004:** Correlation matrix between bone menial density Z-score, sample characteristics, and laboratory results.

	Bone Mineral Density		
	Lumber Vertebrae 1–4	Femoral Neck	Total Body
Weight	0.558 **	0.469 *	0.511 **
Height	0.574 **	0.387 *	0.392 *
BMI	0.568 **	0.447 *	0.505 **
Age	−0.042	0.101	0.095
Ca	−0.026	0.191	0.206
TSH-3	−0.170	−0.164	−0.074
FT4	0.104	0.157	0.122
Vit. D	−0.131	−0.001	0.013

Correlations tested using Pearson’s r. ** Statistically significant, *p* < 0.01. * Statistically significant, *p* < 0.05.

## Data Availability

Data are available upon request.

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
