# Peer review of "The Preventable Effect of Taekwondo Sport among Cadets and Junior’ Bone Mineral Density: DEXA Assessment"

_children, 2023, doi:10.3390/children10010170_

Round 1

Reviewer 1 Report

Hello, the article is interesting, especially since physical activity has special implications in children's lives and is one of the causes of reducing obesity and cardiovascular diseases and other pathologies. in order to be able to move forward with a view to publishing the article, it will be necessary to improve this article. I propose to reanalyze a comparison of BMD and values ​​determined in a control group (same age category, without sport activities, or other group at risk),  or to identify the 95th percentiles in the corresponding age categories. only then will we be able to pass a stage that will allow us to do a systematized analysis and that will bring scientific evidence. I am therefore waiting for a manuscript with a new comparative analysis and please redo what you sent. I wish you best of luck!

Reviewer 2 Report

The article "The Preventable Effect Of Taekwondo Sport Among Cadets And Junior’ Bone Mineral Density: DEXA Assessment" is a descriptive case series study that aims to shed light on the impact of rapid weight loss on BMD among cadets and juniors TKD athletes.

Nevertheless, the paper is difficult to follow since there are no concluding sentences and subtitles on each sections. Presentation is chaotic.

Results are overinterpreted and Conclusions (for example: line 289 "Additionally, taekwondo sport and exercise showed a protector effect for bone density." are barely supported by the Results.

Round 2

Reviewer 1 Report

Hello,I expect you to change the statistics and to be able to demonstrate, based on the results obtained, that this article can be published. at this moment there are only opinions that cannot be supported with results and published. Please add a group of children of similar age and compare the results, so you can then explain what you observed. a scientific article is based on this. as it is now, it is not relevant, I wish you success!

Author Response

Again, we would like to thank you for taking more time to review our paper. however, we would like to highlight a few issues and we would really be grateful if you support our vision. It is ethically unjustified to scan normal children subject of similar age to DEXA scans. It is unlikely the children’s parents or IRB will agree to such scanning without justification. It is extremely rare to do these scans in healthy children. We only have scans in children with known bone diseases, such as osteogenesis imperfecta or Rickets. Healthy Jordanian non-athlete children/young adult reference is essentially impossible to find. We don't do DEXA scans on healthy children. Hence, The z score compares with age- and sex-matched populations. Moreover,  we asked the DEXA company/softwares that report the t and z scores in the DEXA results sheet to have this data but  they answered that it is the t and z scores of the results.

To describe the BMD measurements are first converted into Z‐scores. Z‐scores are similar to T‐scores except that instead of comparing the patient's BMD with the young adult mean, it is compared with the mean BMD expected for the patient's peers (for example, for a healthy normal subject matched for age, gender and ethnic group which are already calculated in the DEXA scan software). Therefore it was not necessary to recruit a control group for comparing the participants results. Hence the reference values of DEXA scan compare the results with matched age and sex from healthy population. Over the last 10 years the interpretation of DXA scans has been guided by the WHO T‐score definition of bone diseases.

Reviewer 2 Report

Thank you very much for the improvements in the Methods part.

English proofreading should be applied. 

Author Response

Dear;

We are really grateful for your positive feedback and will ask the journal for the English language improvement service. 

Thank you again for your time and effort.